# Frost Resistance Investigation of Fiber-Doped Cementitious Composites

**DOI:** 10.3390/ma15062226

**Published:** 2022-03-17

**Authors:** Yongcheng Ji, Yunfei Zou, Yulong Ma, Haoxiang Wang, Wei Li, Wenyuan Xu

**Affiliations:** School of Civil Engineering, Northeast Forestry University, Harbin 150040, China; yongchengji@126.com (Y.J.); zyf13981147047@163.com (Y.Z.); myl2001928@163.com (Y.M.); a1271959050@163.com (H.W.)

**Keywords:** fiber, cementitious composites, frost resistance, numerical simulation, finite element simulation

## Abstract

Fibers used as reinforcement can increase the mechanical characteristics of engineering cementitious composites (ECC), but their frost resistance has received less attention. The mechanical properties of various fiber cementitious materials under the dual factors of freeze-thaw action and fiber dose are yet to be determined. This study examines the performance change patterns of cementitious composites, which contain carbon fiber, glass fiber, and polyvinyl alcohol (PVA) fiber at 0%, 0.5%, and 1% volume admixture in freeze-thaw tests. Three fiber cement-based materials are selected to do the compression and bending testing, and ABAQUS finite element modeling is used to assess the performance of fiber cement-based composite materials. The microscopic observation results show that the dispersion of glass and PVA fibers is higher than that of carbon fibers. As a result, the mechanical characteristics of the fiber-doped cementitious composites increase dramatically after freeze-thaw with increasing dosage. The compression test results show the frost resistance of carbon fiber > PVA fiber > glass fiber. In addition, the bending test results show the frost resistance of carbon fiber > glass fiber > PVA fiber. The 3D surface plots of the strength changes are established to observe the mechanical property changes under the coupling effect of admixture and freeze-thaw times. ABAQUS modeling is used to predict the strength of the cementitious composites under various admixtures and freeze-thaw cycles. The bending strength numerical equation is presented, and the bending and compressive strengths of three different fiber-cement matrix materials are accurately predicted.

## 1. Introduction

Engineering fiber-reinforced cement-based composites (ECC) are used to improve the initial mechanical performance of traditional cement-based materials by doping fibers to overcome the inadequacies of traditional cement-based materials. The impact of various types of fibers (carbon fiber, polypropylene fiber, polyvinyl alcohol fiber, steel fiber, and natural fiber) on the performance of cement-based composites are now the focus of the main study under various doses, lengths, and modification treatments. Furthermore, a few types of research focus on the ideal mix ratio by examining various fiber kinds and sizes. Other studies are focused on using fiber to improve the durability of cement-based products, such as fire resistance, high-temperature resistance, and frost resistance.

To investigate various fibers on the performance of cement-based materials, some scholars primarily conduct mechanical experiments to study the performance improvement of various fiber types, dosages, length dimensions, and modification treatments [1,2,3]. Niu et al. [4] conducted mechanical tests on PVA-ECC specimens with varying fiber concentrations. The test result demonstrates that increased PVA content does not improve the compressive strength but dramatically improves bending strength. Carbon fiber cement-based composites perform better in all aspects of engineering [5,6,7]. Sadrolodabaee et al. [8,9] added a textile waste fiber to prepare fiber-reinforced mortar, and higher compressive strength and stiffness are obtained. Han et al. [10,11,12] filled cement mortar with modified carbon fibers, and the addition of carbon fiber improves the compressive strength, tensile strength, and strain capacity and significantly reduces the volume shrinkage during the hardening process. Mathavan et al. [13,14] evaluated fiber cement-based materials’ mechanical properties and durability, including six natural fibers of cotton, wool, silk, linen, nylon, and polyester as additives. After proper solution treatment, it can be used as a strengthening mechanism for mortar. Feng et al. [15] used nanometer calcium carbonate to modify the surface of polypropylene fibers. The fiber’s surface roughness is increased after modification, and dense hydration products are produced around the fiber’s surface so that the chemical bonding and strength of the fiber and the cement-based interface are improved. Curosu et al. [16,17] treated GFRP and PVA fibers with sulfuric acid and hydrochloric acid, respectively. Compared with sulfuric acid, the hydrochloric acid modification balances the fiber and the matrix’s adhesion, making the fiber cement-based material possess prominent superior mechanical properties. A finite element simulation study is carried out on fiber cement-based composites. The finite element software of ABAQUS is used to simulate fiber concrete numerically. Different types of fiber doping have different effects on the failure mode of concrete. The simulated load-displacement relationship is in good agreement with the test results [18,19]. Hussler-Combe et al. [20] considered the nonlinear behavior of cement matrix, fiber material, and binding law. The theoretical basis, particular implementation problems, verification of simple configuration, and finite element simulation of steel fiber cement-based composite experiments are described.

Other scholars’ research discussed the hybrid fiber mix ratio and size in cementitious composites. Zhang et al. [21] studied the influence of the mixing of basalt fiber and carbon fiber on the mechanical properties of cement. The mixing basalt fiber and carbon fiber can significantly improve the compressive strength and splitting tensile strength of cement. Zhang et al. [22] comprehensively investigated the mechanical properties of ECC made of different microfibers (artificial and natural). The effect of fiber hybridization is discussed, including combining different fibers to form synergistic reinforcement materials. Özkan et al. [23] analyzed the doped mortar’s compressive strength, bending strength, tensile strength, and microstructure. Moreover, the best hybrid combination is obtained with 75% PVA + 25% basalt fiber. Halvaei et al. [24] found that carbon fiber is co-doped with chopped carbon fiber. The crack load and bending toughness increase significantly with increased fiber volume content. Song et al. [25] evaluated steel fiber (SF) and carbon fiber (CF) using the hybrid effect index, compressive toughness, and impact toughness of steel fiber reinforced concrete (SFRC).

In addition, some studies found that fibers have a specific effect on the durability of cement-based composites at a certain amount of fiber [26,27,28]. Nam et al. [29,30] tested the PVA fiber within a specific dosage range to improve cement-based composites’ frost resistance. However, excessive amounts of PVA fibers would adversely affect the frost resistance of cement-based composites. Nuaklong et al. [31] experimentally studied the effect of the hybridization of multi-walled carbon nanotubes (MWCNTS) and polypropylene (PP) fibers on the mechanical properties and fire resistance of cement-based materials. The mortar’s strength is evaluated at temperatures below 1000 °C, and the fire resistance is improved by the MWCNTS bridging effect and the melting of PP fibers. Li et al. [32] examined the steel-polyvinyl alcohol (PVA) fiber-calcium carbonate whisker (CW) multi-scale fiber-reinforced cement-based composite material at a high temperature of 900 °C. Steel-polyvinyl alcohol fiber and CW fiber are two types of fiber. The integration of mortar may significantly strengthen the bending and compressive strength of the mortar after high temperatures. The bending and compressive strength initially increase and then decrease with the temperature increase, and the critical temperatures are 200 °C and 400 °C, respectively. Iurie et al. [33] investigated polyethylene with ultra-high molecular weight, aramid, and high modulus polymers. There is no discernible influence of high temperature on the tensile characteristics of aramid and PBO composites. Fiber cement-based materials comprised of ultra-high molecular weight polyethylene fibers, on the other hand, exhibit considerably better multiple fractures and peak ductility at temperatures up to 150 °C. Liu et al. [34] found that fiber reinforcement reduces the softening degree of frozen-thawed soil in an unconstrained state, thereby improving the soil’s strength properties. Li et al. [35] observed that the mechanical properties and freeze-thaw resistance were improved by increasing cement composites’ basalt fiber content. Li et al. [36] presented a reliability calculation model for freeze-thaw damage based on the Miner theory of cumulative fatigue damage. Tao et al. [37] studied that fiber cement composites’ elastic modulus and unconfined compressive strength decreased as the number of freeze-thaw cycles increased. Chai et al. [38] determined that freeze-thaw actions decrease the interfacial adhesion between fibers and matrix, resulting in more fibers being pulled out during bending loads.

Fibers have greatly improved the different properties of cementitious products. However, few studies have compared the properties and finite element simulations of various fiber-cement matrix composites. Therefore, this paper uses several fiber-cement matrix composites to investigate cementitious composites’ mechanical properties and frost resistance. This study aims to investigate the extent of the effect of different fibers on the properties of cementitious materials under the effect of both doping and freeze-thaw factors. The strength under these factors is well predicted by finite element modeling and software fitting of mechanical data equations. Microscopic techniques are used to observe damaged surfaces under freeze-thaw action. The study combines mechanical testing machine tests and computerized ABAQUS finite elements for mechanical tests, a relatively new approach. Strength predictions are made using Origin and Curve 3D software, and the data were fitted to obtain surface equations. As a result, it can effectively predict the flexural strength of the material, which helps to establish theoretical conditions for engineering practice with other fiber cement-based materials and provides a basis for determining the mix ratio of fiber cement-based composites.

## 2. Experimental Programs

### 2.1. Materials Preparation

Fiber cement-based materials mainly included cement, fibers, fine aggregates, water, and additives. The performance of cement-based materials is improved by adding fibers. Cement: Swan brand P·O 42.5-grade cement is used in this study. All indicators of cement meet the requirements of GB 175-2007 ‘General Portland Cement’. The material property of Portland cement is displayed in Table 1. The river sand was selected for sieving with less than 2 mm particle size. The particle size gradation is shown in Table 2, which conforms to the specification ‘Cement Mortar Strength Test Method (ISO) (GB/T 17671-2020)’. A polycarboxylate water reducer was used for admixture to ensure the workability of the fiber cement composite.

Three types of fibers with a length of 10 mm were selected, including carbon fiber, glass fiber, and PVA fiber (Figure 1). The carbon and glass fiber used in this test were the engineering structural reinforcement materials produced by Kaben Company, Tianjin, China, and the PVA fiber was provided by Shanghai Kaiyuan Chemical Technology, Shanghai, China, as shown in Table 3.

### 2.2. Mix Ratio Design and Test Method

#### 2.2.1. Concrete Mix Design

This study investigated different fibers that affect the mechanical characteristics of cement-based composites. The water-cement ratio was 0.5 for this test’s control group and incorporated carbon fiber, glass fiber, and PVA fiber. The experimental test included seven groups of bending and compressive specimens. Each group of specimens had three identical specimens to guarantee data consistency. Table 4 shows the concrete mix design. Three types of fiber (PVA, carbon, and glass fiber) and three volume contents (0%, 0.5%, and 1%) were selected. The number of freeze-thaw cycles were 0, 50, and 100, used as variables in the seven groups to compare and examine the specimen frost resistance. An appropriate water reducing agent was added to the concrete mix design to ensure workability, as shown in Table 4.

#### 2.2.2. Freeze-Thaw Cycling Test

According to the code of ‘Standard for Long-term Performance and Durability Test Methods of Ordinary Concrete (GB/T 50082-2009)’, the freeze-thaw test uses the rapid freeze-thaw method. As shown in Figure 2, the freeze-thaw cycle device used a freeze-thaw cycle machine that meets the current industry standard ‘Concrete Antifreeze Test Equipment (JG/T243)’. Four days before freezing and thawing, the specimens were immersed in the original solution at 15–20 °C. The freeze-thaw cycle time was 2–4 h, and the thawing time was not less than 1/4 of the cycle time. The minimum and maximum temperature in the center of the specimen should be controlled between (−18 ± 2) °C and (5 ± 2) °C. Freeze and thaw transition time was not more than 10 min. The test specimen was immersed into the water solution of the freeze-thaw cycle box and finally was set to 0, 50, and 100 freeze-thaw cycles.

#### 2.2.3. Bending and Compression Tests

The bending and compression tests were carried out according to the ‘Cement Mortar Strength Test Method (ISO) (GB/T 17671-2020)’. The loading speed of the bending tester met 50 N/s ± 10 N/s. The load was evenly distributed along the width of the specimen without generating torsional stress. After the bending strength test was completed, two half-section specimens were taken out, and the compressive strength test was carried out on the side of the half-section specimens. In the compressive test, the load was uniformly distributed at a rate of 2000 N/s ± 200 N/s until failure. The size of the bending test piece was 40 mm (height) × 40 mm (width) × 160 mm (length), and the effective size of the compression test piece was 40 mm (height) × 40 mm (width) × 40 mm (length).

The specimen’s surface was scraped flat after it had been vibrated.The mold was disassembled and placed in water at intervals for curing at 20 ± 1 °C; after 24 h.The bending and compressive tests were carried out.

The YAW-300H testing machine produced by Jinan Hengruijin was used for the test, as shown in Figure 3.

#### 2.2.4. Concrete Microstructure Observation

In order to further study the different fibers’ mechanisms, a sample was selected from the bending test. A GAOSUO digital microscope was used to examine the cross-sections of several fiber cement-based material specimens microscopically (observation magnification was 400 times), as shown in Figure 4. The GAOSUO digital microscope is produced by Xinbaiyi Technology Company in Shenzhen, China.

GAOSUO digital microscope specifications are as follows:Magnification: 1×~500×Imaging distance: manual adjustment 0~infinityImage resolution: standard 640 × 480Fixed base: universal metal base (optional lifting bracket)

## 3. Test Results and Analysis

### 3.1. Surface and Internal Observation after Freezing and Thawing

Each group has three test pieces with the same conditions. Figure 5a,b shows the appearance of the specimens after 0, 50, and 100 freeze-thaw cycles of undoped fiber and fiber-doped cement-based composites, respectively. The red box denotes visual observation (1× magnification for observation). The specimens were observed to have a smooth surface after 0 freeze-thaw cycles. However, after 50 freeze-thaw cycles, the surface was eroded while the material was peeling off. The fiber cement base had a better appearance and integrity than the unmixed cement base, and there were a small number of fiber filaments attached to the surface. After 100 freezes and thaws, it was evident that the surface of the cement-based composite without fibers was severely eroded and dropped many pits. Partial scaling of the concrete surface was observed after 100 freeze-thaw cycles. However, the addition of the fibers prevented scaling damage, thereby preserving the specimen’s integrity. As a result, the surface integrity of the cement-based composite was much better than that of the cement-free composite. The results reveal that the integrity of the cement-based doped fiber is better than the undoped fiber when it comes to frost resistance.

Figure 5c shows the appearance of three fiber cement-based materials after 100 freeze-thaw cycles and the appearance of the fiber under 400× magnification observation. When exposed to freeze-thaw, the white cement matrix on the surface peeled away, revealing the sand and fibers beneath. It can be observed that the carbon fiber color is black, while the glass fiber is white, and both surfaces are smooth. However, the PVA fiber is light yellow, and the surface is rough. As a result, irregular pits appeared on the exposed aggregate’s surface, mainly formed of two colors: black and yellow.

By observing the failure section of the bending test, the carbon fiber cement-based composite material under different freezing and thawing times is shown in Figure 6a. The carbon fiber filaments are found to be black and opaque, with tension fracture-type damage and a slight degree of cement-based slippage. With the increase of freeze-thaw times, the degree of slippage between carbon fiber and cement-based material increased. The glass fiber failure section is shown in Figure 6b. Fiberglass is white, translucent, and reflective. Glass fibers had a more pronounced dispersion effect. Comparing carbon fiber cement-based materials, the slip between the glass fiber and the cement base was more significant, and the failure modes were mostly broken and pulled out. The failure section of the PVA fiber composite is shown in Figure 6c. PVA fiber is yellowish and opaque, and slippage and fracture are the most common failure modes. It can be clearly seen that the sliding portion of the fibers is soft and curled. PVA fiber has good dispersibility in cement foundations without cluster aggregation. By comparison, the dispersion effect of glass fiber in the section of cement-based composite material is better than that of PVA fiber, and PVA fiber is better than carbon fiber. With the increase of freezing and thawing times, the slippage between the three fibers and the cement base becomes more and more apparent. The primary failure mode of the bending test was fiber pull-off at 0 freeze-thaw cycles. After 50 freeze-thaw cycles, slippage and fracture were the most common failure modes. After 100 freeze-thaw cycles, the failure modes were mostly broken and pulled out due to the enormous sliding between the fibers and the cement base. At the same time, it was found that cracks are formed inside the cement base after 100 freeze-thaw cycles.

The glass fiber’s dispersion effect in the cement-based composite material section is better than PVA fiber and carbon fiber. As the number of freezing and thawing increased, the slippage of fibers and cement-based materials increased. However, carbon fibers ensure lower slippage with cement-based materials than other fibers after freezing and thawing. The main reason is that the carbon fiber has a higher elastic modulus and reduced the deformation under freezing and thawing.

### 3.2. Mechanical Strength under Freeze-Thawing Effect

The compression test results are shown in Table 5. ANOVA analysis was performed on the three groups of data using SPSS software, and the significance of the analysis results was significantly less than 0.05. The test achieved statistical significance and was statistically significant. The standard deviation increased with increasing fiber content, indicating that the random distribution of fibers affects individual data value. The compressive strength in the table is the average strength of the three tests. The header row is the different fiber types and content, and the header column is the number of freeze-thaw cycles. The statistical analysis results are shown in Figure 7, which shows compressive strength changes of carbon fiber, glass fiber, and PVA fiber composite material when subject to various freeze-thaw conditions and fiber volume fractions. Specimens with 0, 0.5%, and 1% fiber contents were subjected to 0, 50, and 100 freeze-thaw cycles, respectively. The compressive strength fell as the freeze-thaw periods rose and increased initially, then dropped as the volume fraction increased. It can be seen from Figure 7a that the compressive strength of fiber cement-based composites at 0.5%volume fraction decreases in the order of carbon fiber, PVA fiber, and glass fiber. The maximum strength of 0.5%volume fraction carbon fiber was 46.8 MPa under 0 freeze-thaw cycles. It had the highest strength even after 50 and 100 freeze-thaw cycles, although decreased by 8.1% and 16.7%, respectively. The glass fiber composite strength decreased from 9.5% to 20.4%. The strength of PVA fiber composites decreased by 9.9% and 18.6%. It can be seen from Figure 7b that under 0 freeze-thaw cycles, the strength of the fiber cement-based composite material with a volume fraction of 1% decreased successively, which is the compressive strength of the control group carbon fiber, PVA fiber, and glass fiber. After 50 freeze-thaw cycles, the intensity of the control group dropped sharply. The strength of carbon fiber composites decreased by 3.8% and 9.6% after 50 and 100 freeze-thaw cycles, respectively. Glass fiber composite materials decreased by 6.7% and 11.6%. PVA fiber composite materials decreased by 3.4% and 12.2%. It can be seen from Figure 7c that the compressive strength of the fiber cement-based materials is compared at the three-volume fractions of 0, 0.5%, and 1%. The law of compressive strength increases first and then decreases with the increase of volume fraction, which is not affected by the number of freeze-thaw cycles. The intensities were summed under the same volume fraction of three freeze-thaw cycles. The compressive strength of carbon fiber cement-based composites increased by 12.1% from 0% to 0.5% volume fraction but decreased by 7.8% from 0.5% to 1%. Glass fiber cement-based composites increased by 8.3% and decreased by 11.6%. PVA fiber cement-based composites rose by 4.2% and fell by 12.3%. This shows that carbon fiber has the most noticeable impact on the compressive strength of cement and materials, followed by PVA fiber and glass fiber.

Compressive strength can be increased by incorporating a certain volume percentage of fibers, but it will significantly decrease beyond a limit percentage. It can be explained that fibers in cement-based materials have a unique bridging effect, limiting the expansion range of internal cracks. Uneven dispersion of internal fiber clusters and voids will result from an excessive volume fraction. On the other hand, an increase in voids degrades the cement body’s compactness and strength. First, different fibers in cement have a bridging effect determined by their elastic modulus and tensile strength. Second, the effect is proportional to the friction force between the cement bases. Carbon fiber’s elastic modulus is significantly greater than that of glass and PVA fiber, making it the best material for this application. A microscopic examination revealed that PVA fiber had the highest dispersibility and exhibited no slippage when mixed with cement. As a result, carbon fiber has a more significant reinforcing effect at a specific dosage than PVA or glass fiber. The compressive strength of fiber-doped cement-based materials was substantially more significant than that of the control group after 50 and 100 freeze-thaw cycles. This demonstrates that the fiber’s frost resistance to the cement-based composite material has increased dramatically. The rationale for this is that fibers can help prevent fractures in cement-based composites induced by thermal expansion and contraction during freeze-thaw. Additionally, the fiber cement composite expands and contracts continuously and provides energy for the fiber with increasing freezing and thawing times. As a result, the absorbed energy of fiber reaches a limit level, losing the enhancement effect. This demonstrates that carbon fibers store the most energy of the three fibers and withstand greater loads. It can be determined that the strength of carbon fiber cement-based composites declines the slowest with increasing freeze-thaw periods, followed by PVA fiber and glass fiber, at 0.5% and 1% volume content. Carbon fibers have a high modulus of elasticity (Table 3) and relatively minimal mortar slippage was observed by microscopy. Therefore, based on the compression test and microscopy observations, it seems that the frost resistance of fiber reinforced concrete mixtures evolved according to this sequence; carbon fiber > PVA fiber > glass fiber.

Table 6 shows the results of the bending test. The bending strength in the table is the average strength of three tests. The header row is the different fiber types and contents, and the header column is the number of freeze-thaw cycles. The statistical analysis results of the data in Table 5 are shown in Figure 8, which shows the change in compressive strength of carbon fiber, glass fiber, and PVA fiber composites when subjected to various freeze-thaw conditions and fiber volume fractions. Specimens with 0, 0.5%, and 1% fiber content were subjected to 0, 50, and 100 freeze-thaw cycles, respectively. Shown in Figure 8a,b are the bending strength changes of carbon fiber, glass fiber, and PVA fiber under different freeze-thaw times and volume fractions. Figure 8c is a strength histogram. The bending strength dropped as the freeze-thaw duration increased, but the strength rose as the volume fractions grew. It can be seen from Figure 8a,b that the bending strength of carbon fiber composites is the highest. Figure 8a shows that the greatest strength for a 0.5% volume fraction of carbon fiber composite material under 0 freeze-thaw cycles is 7.6 MPa. Carbon fiber composites had the highest strength even after 50 and 100 freeze-thaw cycles. Glass fiber and PVA fiber composites have comparable bending strengths. The strength of the carbon fiber composites decreased by 3.9% and 7.9% after 50 and 100 freeze-thaw cycles, respectively. The strength of glass fiber composites decreased by 3.1% and 6.3%. The strength of the PVA fiber composite material decreased by 4.6% and 7.8%. It can be seen from Figure 8b that the highest strength is 7.6 MPa for a 1% volume fraction of carbon fiber composite material under 0 freeze-thaw cycles. It had the highest strength even after 50 and 100 freeze-thaw cycles. The strength of the carbon fiber composites decreased by 3.9% and 7.9% after 50 and 100 freeze-thaw cycles, respectively. The strength of glass fiber composite material decreased by 1.5%. The strength of the PVA fiber composite material decreased by 3.1% and 6.2%. The bending strength of the three fiber cement-based materials is compared in Figure 8c at three-volume fractions of 0, 0.5%, and 1%. With the increase of volume fraction, the growth rate of bending strength was rapid at first and gradually became gradual, which was unaffected by the number of freeze-thaw cycles. The intensities were summed under the same volume fraction of three freeze-thaw cycles. The bending strength of carbon fiber cementitious materials increased by 47.3% from 0% to 0.5% volume fraction and by 11.7% from 0.5% to 1% volume fraction. The bending strength of glass fiber cement-based materials improved by 39.8% and 4.8%. The bending strength of PVA fiber cement-based materials rose by 38.3% from 0 to 0.5% volume fraction, but, from 0.5% to 1% volume fraction, it declined by 2.7%. It shows that carbon fiber has the most prominent influence on the bending strength of cement and materials, followed by glass fiber and PVA fiber.

The test suggests that fiber doping can boost bending strength considerably. This is due to the fiber’s unique bridging properties within the cement-based substance. The bottom of the cement-based composite material gets stretched when the higher end is crushed. Because the cement-based substance has a low tensile strength, it can help to prevent bottom fractures from developing. The bending strength of the fiber-doped cement-based material was considerably higher than that of the control group after 50 and 100 freeze-thaw cycles, demonstrating that the fiber is helpful to the cement-based composite material’s frost resistance. Carbon fiber had the most prominent influence on the bending strength of cement and materials, followed by glass fiber and PVA fiber. It can be concluded that the strength of the carbon fiber cementitious material decreases the slowest with an increasing number of freeze-thaw cycles, followed by PVA fiber and glass fiber. Carbon fibers have a high modulus of elasticity that helps cement composites resist deformation and crack formation during freeze-thaw cycles. In the bending test, the mortar’s tensile strength and ultimate strain were small. The bending strength mainly depends on the fiber bearing capacity, and the effect of the fiber is noticeable. Carbon fibers with higher tensile strength and less elongation had the greater load-bearing capacity in bending tests. Additionally, carbon fibers store more energy than other fibers, protecting them from premature failure during freeze-thaw cycles. Thus, carbon fiber exerts a better frost resistance than other fibers in the bending test, followed by glass and PVA fiber.

### 3.3. Bending-Compressive Strength Ratio

The bending-compressive strength ratio can be used to reflect the crack resistance of the material, and it can also characterize the ductility and toughness of cement-based composites. Under the circumstances of 0, 50, and 100 times freeze-thaw, the bending-compressive strength ratio of PVA, glass, and carbon fiber composite materials is determined at 0%, 0.5%, and 1% volume fraction, respectively. The bending-compressive strength ratio results are shown in Table 7, and Figure 9 shows the stacking histogram of the bending-compressive strength ratio of the fiber cement-based material with the same content and type. The bending-compressive strength ratio rises typically as the number of freeze-thaw cycles rises and as the dose rises.

The bending-compressive strength ratio of 1% carbon fiber composite material was as high as 18.25% without freeze-thaw cycles. When the number of freeze-thaw cycles was 50, the bending-compressive strength ratio of 1% glass fiber composites was as high as 19.1%. The maximum bending-compressive strength ratio of 1% volume fraction of carbon fiber composites was 19.08% under the action of 100 freeze-thaw cycles. Compared with the control group, the bending-compressive strength ratio of cement-based composites continued to increase with fiber content, and 1% was more significant than 0.5% and greater than 0. The three fiber composite materials’ bending-compressive strength ratio showed an upward trend. PVA fiber composite materials increased marginally, whereas carbon fiber and glass fiber composite materials significantly increased.

The fiber volume is positively related to the more excellent bending-compressive strength ratio. As a result, fiber can significantly increase the sample’s toughness and ductility. The bending-compressive strength ratio is influenced differently depending on the kind of fiber, with glass fiber being greater than carbon fiber and PVA fiber. The impacts of freeze-thaw will exacerbate this discrepancy.

### 3.4. Strength under Volume Fraction and Freeze-Thawing Coupled Effect

#### 3.4.1. Space Surface Diagram of Compressive Strength

The 3D surface maps for different fiber types are established for fiber cement-based composites under the two variables of freeze-thaw duration and dose. For example, Figure 10 represents three fiber composite materials: carbon fiber, glass fiber, and PVA fiber, respectively. In the curved surface graph, the compressive strength of the three fiber composites is the highest at 0.5% volume fraction and 0 freeze-thaw cycles, which are 46.8 MPa, 44 MPa, and 45.5 MPa, respectively. Therefore, it presents a triangular pyramid shape in space.

The three fiber composites’ compressive strength grew initially and declined when the freeze-thaw cycles remained constant. The compressive strength diminished as the number of freeze-thaw cycles rose while the fiber content remained constant. The shaded portion of the projected area at 0.5% volume fraction and 0 freeze-thaw cycles was observed. The strength of the doped carbon fiber composites decreased toward the periphery and were relatively symmetrical at about 0.5% volume fraction. The difference in compressive strength between 0 percent and 0.5 percent volume fractions was not apparent in the glass fiber composite, and the influence of the dose was modest. However, the strength dropped sharply when the dosage was more significant than 0.5%. The strength of the glass fiber composite material reached the lowest when the freeze-thaw cycle was 100 times and the volume fraction was 1%. The change rule of PVA fiber composite material in the figure was similar to that of carbon fiber composite material. The two fiber composite materials were symmetrical at about 0.5%, but the compressive strength of PVA fiber composite materials was lower than that of carbon fiber at any point. Coordinate points (x, y, z1) signify the following: volume fraction, freeze-thaw times, compressive strength. The highest point of carbon fiber composite materials reached the maximum compressive strength at O1 (0.5, 0, 46.8), while the lowest point reached the lowest compressive strength at P1 (0, 100, 41.6). The highest point of glass fiber composite material reached the maximum compressive strength at O2 (0.5, 0, 44), while the lowest point reached the lowest compressive strength at P2 (1, 100, 32.7). The highest point of PVA fiber composite material reached the maximum compressive strength at O3 (0.5, 0, 45.5), while the lowest point reached the lowest compressive strength at P3 (1, 100, 33.7). The amount and number of freeze-thaw cycles corresponding to the highest strength point were the same for the three fiber composites, but the amount and number of freeze-thaw cycles corresponding to the lowest strength point were different.

The appropriate amount of fiber cement-based composite material can significantly improve its frost resistance, thereby increasing the compressive strength of the cement-based material. However, excessive fiber will result in insufficient internal compactness of cement-based materials and an increase in voids, which will cause the frost resistance to decrease again. In addition, the compressive strength decreases with the increase in the number of freeze-thaw cycles. The compressive strength was the highest when the dosage was 0.5%, and the freeze-thaw cycle was 0 times. Therefore, the compressive strength presents a triangular pyramid-shaped space surface.

#### 3.4.2. Space Surface Diagram of Bending Strength

A 3D space map was created for several fiber types in fiber cement-based composites under two variables: freeze-thaw timeframes and content. Figure 11 represents three fiber composite materials: carbon fiber, glass fiber, and PVA fiber. The maximum bending strengths of the three fiber composites were 7.6 MPa, 6.7 MPa, and 6.5 MPa, respectively. The volume fraction was 1%, with 0 freeze-thaw cycles showing a bending line in the surface plot.

The bending strength of the three fiber composites increased sharply from 0 to 0.5% by volume fraction when the number of freeze-thaw cycles was constant. However, the bending strength increased slowly between 0.5% and 1% by volume fraction. The bending strength decreased slowly as the number of freeze-thaw cycles increased as the fiber content was constant. It shows that the fiber can improve the bending strength of cement-based materials after freeze-thaw. The bending strength has a tilted contour map distribution when seen from the perspective of the 3D projection surface. Coordinate points x, y, and z2 respectively represent dosage, freeze-thaw times, and bending strength. The highest point of carbon fiber composite materials reached the maximum bending strength at M1 (1, 0, 7.6), and the lowest point reached the lowest bending strength at N1 (0, 100, 4.1). The highest point of glass fiber composite material reached the maximum bending strength at M2 (1, 0, 6.7), and the lowest point reached the lowest at N2 (0, 100, 4.1). The highest point of PVA fiber composite material reached the maximum bending strength at M3 (1, 0, 6.5), and the lowest point reached the lowest bending strength at N3 (0, 100, 4.1). The volume fractions and freeze-thaw times corresponding to the highest and lowest points of the three fiber composite materials were the same. The more fiber content, the greater the bending strength, and the more freeze-thaw cycles, the lower the bending strength.

The test results show that the fiber works when the cement-based material is bent at a low fiber content, preventing bottom cracks, thereby significantly improving the bending strength. However, if the volume fraction exceeded 0.5%, the material’s bending strength decreased. At the same time, the bending strength decreased with the increase of freeze-thaw times. Therefore, the bending strength reaches its highest when the volume fraction is 1% with 0 freeze-thaw cycles and exhibits a spatially folded surface. Carbon fiber, glass fiber, and PVA fiber composites exhibit relatively consistent patterns with different strength magnitudes. This has to do with the fiber’s physical characteristics and its interaction with cement-based materials.

## 4. Numerical Simulation Analysis

The bending strength of fiber cement-based composites exhibits the same equation under the action of different fiber volume fractions and freeze-thaw cycles. On this foundation, a formula model for the bending strength of cement-based composites with different fiber doping under the action of volume fraction and freeze-thaw is proposed.

After determining the type of fiber, we request to optimize a performance parameter or multiple performance parameters. We present a mathematical method for this type of binary parameter optimization. For practical problems, a particular performance parameter usually first increases and then decreases with the increase of the independent variable (or first decreases and then increases). Therefore, we assume that this is a binary function describing the problem. Figure 11 is mathematically modeled using Origin software, which automatically selects the fitted surface to obtain Equation (1). It shows the generic form, which has six unknowns. Solving these six unknowns requires six independent conditions of x, y, and z. These six parameters are replaced into the equation once solved (1). When x = 1 and y = 0, there is a maximum value in the function range. If x = 0 and y = 100, the function range has a minimum value.
z = z_0_ + B × pow(x,c) + D × pow(y,E) + F × pow(x,C) × pow(y,E)(1)
where z is bending strength; z_0_ is 0% volume fraction and 0 freeze-thaw bending strength; B, C, D, E and F are constant parameters; x is fiber volume fraction; y is freeze-thaw times. The main function of the POWER function is to return the power of a given number. Parameter a represents the base, and parameter b represents the exponent.

The six parameters of various fibers are shown in Table 8. The fixed constant z_0_, for example, indicates the bending strength under zero content and zero freeze-thaw cycles. The parameter values B, C, D, E, and F impact the bending strength of various fibers under varied content x and freeze-thaw periods y. The greater the value of B, C, F, F parameters, the greater the strength, indicating the better the material properties. There is no apparent link between the size of C and D and the strength.
z = 4.8 + 2.97 × pow(x,0.5) − 0.03 × pow(y,0.69) + 0.00053 × pow(x,0.5) × pow(y,0.69)(2)
z = 4.8 + 1.95 × pow(x,0.22) − 0.004 × pow(y,1.107) + 0.0015 × pow(x,0.22) × pow(y,1.107)(3)
z = 4.8 + 1.73 × pow(x,0.14) − 0.02 × pow(y,0.82) + 0.006 × pow(x,0.14) × pow(y,0.82)(4)

Figure 12 shows the simulated formula model presents a curved “waterfall” state, a paraboloid with a downward opening. The maximal point of the paraboloid is y = 0, and it is at x = 1. X is in the range of 0–0.5, the intensity increases rapidly, and there is a clear change trend. The intensity drops dramatically in the 0.5–1 interval. After prediction, the quantity of fiber will no longer increase the bending strength of cement-based composites. Y is in the range of 0–50. The larger of y, the smaller the intensity. It is predicted that as the number of freeze-thaw cycles y increases, the final strength will drop to zero. The coordinate points x, y, z denote the following: dosage, freeze-thaw times, bending strength. Figure 12a is the bending strength model of glass fiber cement-based composite material under the action of the dosage and the number of freeze-thaw cycles. We choose the three most unique places, a1, b1, and c1, which indicate the maximum bending strength, the turning point of strength shift, and the lowest bending strength, respectively. The positions of the three points are a1 (1, 0, 7.7), b1 (0.5, 0, 6.9), and c1 (0, 100, 4.1). Figure 12b shows the bending strength model of the carbon fiber cement-based composite material under the action of the dosage and the number of freeze-thaw cycles. We choose three of the most characteristic points, a2, b2, and c2, representing the highest point of bending strength, the turning point of strength change, and the lowest point of bending strength. The positions of the three points are a2 (1, 0, 6.75), b2 (0.5, 0, 6.47), and c2 (0, 100, 4.14). Figure 12c is the bending strength model of PVA fiber cement-based composite material under the action of the dosage and the number of freeze-thaw cycles. We choose three of the most characteristic points, a3, b3, and c3, representing the highest point of bending strength, the turning point of strength change, and the lowest point of bending strength. The positions of the three points are a3 (1, 0, 6.5), b3 (0.5, 0, 6.37), and c3 (0, 100, 4.0).

The optimal volume fraction may be estimated using this experimental approach under the known freeze-thaw periods of a specific fiber composite material. This method verifies that the fiber improves cement-based materials’ mechanical properties and frost resistance. It can reveal the law of bending strength of different kinds of fiber cement-based materials with the number of freeze-thaw cycles and volume fraction. This approach is appropriate since the strength of cement-based materials is discrete. The equation’s freeze-thaw times and mixing amount should be selected according to particular engineering experience, and the corresponding parameters have the correct range of variation.

## 5. Finite Element Modelling

Firstly, the simulation utilizes Python to generate fiber code. The principle of the Python code is first to determine the three-dimensional parameters of the specimen, such as 4*4*16 cm, and then import the fiber diameter, length, and volume content. The coordinates of the fibers appeared randomly within the three-dimensional parameters of the specimen. It was imported into ABAQUS to build uniformly distributed and discrete fiber parts. Then, concrete parts were built, and the C40 concrete damage plasticity parameters were given to the cement-based composites under 0, 50, and 100 freeze-thaw cycles. Moreover, the performance indicators such as elasticity, plasticity, and density of the glass fiber were given to the fiber. Finally, the contact between the fibers and the cement-based material was done in a “built-in” fashion. The mesh accuracy of the fiber part is four times that of the concrete mesh. The composite model was subjected to multiple compression and bending simulation tests, and the following results were finally obtained.

### 5.1. Stress Simulation Analysis

A 4 cm cube fiber cement-based composite was obtained by ABAQUS modeling to simulate the compression test. Compared with Figure 13a, the ABAQUS simulated stress cloud diagram in Figure 13b shows that the stresses on the upper and lower sides are relatively concentrated, and the surrounding stresses are relatively small, which is in line with the experimental results. Figure 13c shows the ABAQUS simulated stress perspective cloud diagram, showing the stress relationship between internal fibers and external cement-based materials. The fiber is less stressed, and some are not stressed, as seen in the partially enlarged picture. Comparative analysis shows that the compression of fiber cement-based composites mainly occurs when the internal cement matrix is damaged. The maximum failure stress is 36 MPa when the downward displacement of the top model reaches the maximum displacement of the measured test.

A 4 × 4 × 16 cm fiber–cement composite was created using ABAQUS modeling to simulate a bending test, where two arcuate rigid bodies support the bottom and the upper half is the test piece. The ABAQUS simulated stress cloud diagram shows that stress concentration occurs at the two points at the bottom and the contact point where the force is applied at the top, as shown in Figure 14a,b. Tensile stress appears at the bottom of the fiber cement base, reaching about 2 MPa. Figure 14c shows the ABAQUS simulated stress perspective cloud diagram, showing the comparison of stress cloud diagrams between internal fibers and external cement-based materials. The stress of the fiber at the section increases from top to bottom. The non-section fibers do not participate in the work. Through comparative analysis, it is found that the fiber cement-based composite material is fractured mainly at the bottom of the cement matrix. The fibers at the section play a role in hindering the development of cracks. The farther to the bottom, the greater the stress and the more severe the damage. The maximum stress at failure is 7.0 MPa when the downward displacement of the top model reaches the maximum displacement of the measured test.

### 5.2. Comparison between Simulated and Experimental Test Results

Figure 15 shows the maximum load of the three fibers. The test and simulation comparison chart can clearly and intuitively express the changing law of the ultimate load of the specimen under different conditions. The comparative finite element simulation and bending test results of three fiber cement-based materials are shown in Figure 15a,c,e, respectively. With increasing volume percent, both measured and simulated bending strength increased. The bending strength decreased with the increase of freeze-thaw times. The measured bending strength is less than the simulated bending strength. The ultimate measured load of the unfreeze-thaw cement-based composite material under 1% carbon fiber volume fraction reached a maximum of 3.236 KN, and the ultimate simulated load can reach 3.317 KN. Under 0.5% fiber volume fraction and 100 times freeze-thaw cement-based composites, the ultimate measured load reached as low as 2.635 KN, and the ultimate simulated load can reach 2.719 KN. For fiber cement-based composites under the action of 100 freeze-thaw cycles, the reduction rate of bending strength of 1% volume fraction carbon fiber cement-based materials is significantly higher than that of 0.5% volume fraction carbon fiber cement-based materials. The other two kinds of fibers also exhibit this law. The ultimate measured load of the unfreeze-thaw cement-based composite material under 1% volume fraction of glass fiber reached 2.847 KN, and the ultimate simulated load can reach 2.911 KN. The ultimate measured load of 100 freeze-thaw cement-based composites under 0.5% glass fiber volume fraction reached as low as 2.55 KN, and the ultimate simulated load can reach 2.636 KN. The ultimate measured load of the unfreeze-thaw cement-based composite material of PVA fiber under 1% volume fraction reached 2.762 KN, and the ultimate simulated load can reach 2.844 KN. The ultimate measured load of 100 freeze-thaw cement-based composites under 0.5% PVA fiber volume fraction reached as low as 2.507 KN, and the ultimate simulated load can reach 2.585 KN. The law of the limit load of the simulation test results with the volume percent and the number of freeze-thaw cycles is determined to be compatible with the actual test results after comparing the actual test and simulation results. After comparing the actual test and simulation results, the simulation results of the three fiber cement-based materials all show that as the volume fraction increases, the ultimate load first increases rapidly and then tends to relax. The ultimate bending load decreases as the number of freeze-thaw cycles increases. Frost resistance can be improved by increasing the fiber volume percentage. The law of the ultimate load of the bending simulation test results with the volume fraction, and the number of freeze-thaw cycles is consistent with the actual test results.

Figure 15b,d,f are the comparative finite element simulation and compression test results of three fiber cement-based materials, respectively. The measured compressive strength and simulated compressive strength decrease with the dosage increase, and the compressive strength decreases with the increase of freeze-thaw times. As a result, the measured compressive strength is lower than that predicted by the simulation. Among them, the ultimate measured load of the unfreeze-thaw cement-based composite material with 0.5% carbon fiber content reached as high as 74.8 KN, and the ultimate simulated load can reach 77.1 KN. The ultimate measured load of 100 freeze-thaw cement-based composites with 0.5% carbon fiber content reached as low as 59.5 KN, and the ultimate simulated load can reach 62.2 KN. For fiber composite materials under 100 freeze-thaw cycles, the ultimate load of 1% carbon fiber cement-based materials is significantly higher than that of 0.5% carbon fiber cement-based materials. Glass fiber also exhibits this law. The ultimate measured load of the unfreeze-thaw cement-based composite material at 0.5% of the glass fiber content reached 72.8 KN, and the ultimate simulated load can reach 74.1 KN. The ultimate measured load of 100 freeze-thaw cement-based composites with 0.5% fiber content reached as low as 53.9 KN, and the ultimate simulated load can reach 55.2 KN. The ultimate measured load of the unfreeze-thaw cement-based composite material with PVA fiber at 1% content reached 70.4 KN, and the ultimate simulated load can reach 72 KN. The ultimate measured load of 100 freeze-thaw cement-based composites with 1% PVA fiber content reached as low as 52.3 KN, and the ultimate simulated load can reach 54.1 KN. After comparing the actual test and simulation results, all three fiber cement-based materials’ simulation results show that as the volume fraction increases, the ultimate load first increases and then decreases. The ultimate compressive load gradually decreases as the number of freeze-thaw cycles increases. Therefore, frost resistance can be improved by increasing the fiber volume percentage. The law of the ultimate load of the compression simulation test results with the volume fraction and freeze-thaw times is consistent with the actual test results.

It can be concluded that the fibers present a uniform and non-cross distribution under the simulation inside the cement base. However, the measured strength is somewhat lower than the predicted value due to irregular local fiber dispersion during the actual test. The findings of the finite element modeling and the actual test results show that the fiber may improve the mechanical characteristics of cement-based composites after freeze-thaw, regardless of whether the test is bending or compressive. Through the established finite element model, the compressive and bending strengths of carbon fiber cement-based materials, glass fiber cement-based composite materials, and PVA fiber cement-based composite materials can be predicted within the range of 1% volume fraction and 100 freeze-thaw cycles. For example, the bending strength of carbon fiber cement-based materials can be predicted at a volume fraction of 0.6% after 30 freeze-thaw cycles. The strength of fiber cement-based materials is successfully estimated with a volume percentage of more than 1% and 100 freeze-thaw cycles.

### 5.3. Correlation Analysis

This paper uses the well-known ‘Pearson’s correlation coefficient’ to study the relationship between strength, freeze-thaw cycles, and volume percent obtained under simulated experiments. The Pearson correlation coefficient is often used in the natural sciences to evaluate the degree of connection between two variables, and its value ranges from −1 to 1. The calculation Formula (5) is as follows:(5)$$r=\frac{\sum_{i=1}^{n}\left(X_{i}-\bar{X}\right)\left(Y_{i}-\bar{Y}\right)}{\sqrt{\sum_{i=1}^{n}{\left(X_{i}-\bar{X}\right)}^{2}}\sqrt{\sum_{i=1}^{n}{\left(Y_{i}-\bar{Y}\right)}^{2}}}$$
where X_i_ is sample mean (i = 1, 2, 3…); $\bar{X}$ is number of samples; Y_i_ is sample mean (i = 1, 2, 3…); $\bar{Y}$ is number of samples.

Four columns of data include fiber content, number of freeze-thaw cycles, and simulated flexural and compressive strengths for each set of samples. The determined fiber content and the number of freeze-thaw cycles determine the flexural and compressive strengths. In addition, Pearson’s correlation analysis was performed on the four columns of data. Finally, all data in Table 9, Table 10 and Table 11 are obtained. The purpose is to study the correlation between different fibers’ flexural and compressive strengths. This is consistent with the research direction of dosage and freeze-thaw times on the strength of different fiber cement-based materials.

This paper studies the strength under the two factors of dosage and freeze-thaw times. The columns and rows represent the correlation between the fiber content and the number of freeze-thaw factors and the strength. The Pearson correlation coefficient of carbon fiber cement-based composite materials is shown in Table 8, consisting primarily of compressive strength, bending strength, volume percentage, and freeze-thaw durations. It can be seen from Table 9 that the compressive strength of carbon fiber cement-based composites is positively correlated with the bending strength and volume parameters, but the correlation is not high. Its compressive strength is negatively correlated with the number of freeze-thaw cycles, and the correlation is 0.05, which is significant. The bending strength of carbon fiber cement-based composites is negatively correlated with the number of freeze-thaw cycles, but the correlation is not high. Its bending strength is positively correlated with volume fraction, the correlation is 0.01, and the correlation is significant. The number of freeze-thaw cycles affects the compressive strength of carbon fiber cement-based materials the most, whereas the integral number of receptors affects the bending strength the most. Table 10 displays the Pearson’s prior relation coefficients of glass fiber cement-based composite materials, mainly composed of the compressive strength, bending strength, volume fraction, and freeze-thaw times of glass fiber cement-based materials. It can be seen from Table 9 that the compressive strength of glass fiber cement-based composites is positively correlated with the bending strength and volume parameters, but the correlation is not high. The number of freeze-thaw cycles is inversely connected with compressive strength, and the association is significant at the 0.01 level. Its compressive strength is negatively correlated with volume fraction, and the correlation is not high. The bending strength of glass fiber cement-based composites is negatively correlated with the number of freeze-thaw cycles, but the correlation is not high. Its bending strength is positively correlated with volume fraction, the correlation is 0.01 level, and the correlation is significant. Table 11 shows the Pearson correlation coefficient of PVA fiber cement-based composite materials. The four metrics of compressive strength, bending strength, volume fraction, and freeze-thaw periods are all in good agreement with glass fiber cement-based composites.

It is concluded that the compressive strength of the three fiber cement-based materials have a significant correlation with the number of freeze-thaw cycles, and the bending strength has a significant correlation with the volume fraction. However, the Pearson coefficients of the three fibers are different. Therefore, the fiber has different effects in the compression and bending tests and can play a better role in the bending test. Therefore, the dosage correlates with the flexural strength and a low correlation with the compressive strength. The freeze-thaw effect is mainly on the cement base, so the freeze-thaw has a high correlation with the compressive strength and a low correlation with the flexural strength.

### 5.4. Strength Prediction Model under Different Fiber Dosage

The two factors are stacked to provide a comprehensive strength prediction method that predicts the 0 percent to 1 percent volume fraction and strength under 0 to 100 freeze-thaw cycles. Each model is calculated once for the fiber content of 0%, 0.5%, and 1% and freeze-thaw cycles of 0, 50, and 100 times, yielding nine strength values that linearly regress to a surface. The fitting was performed with the formula of Curve 3D software itself, and the formula of the fitting surface with the minimum error was obtained, as shown in Equation (6):(6)$$z={a+\mathrm{bx}+\mathrm{cy}+\mathrm{dx}}^{2}{+\mathrm{ey}}^{2}+\mathrm{fxy}$$
where z is strength; a, b, c, d, e, and f are constant parameters; x is fiber volume fraction; y is freeze-thaw times.

Figure 16 shows the strength prediction model of carbon fiber cement-based composites. Figure 16a shows the compressive strength prediction model of carbon fiber cement-based materials, which presents a “sandpile”-like curved surface. In order to find the extreme value of Formula (5), firstly substitute the parameters in Figure 16a into Formula (5) to obtain the corresponding formula, and then calculate the partial derivative of x and y corresponding to formula z. Solving the equations yields x = 0, y = 0.5, and z = 46.28. At this time, the carbon fiber cement-based polymer has reached its maximum compressive strength of 46.28MPa. The equation is solved to obtain x = 0, y = 1, and z = 7.62, as illustrated in Figure 16b. The carbon fiber cement-based composite reaches its maximum bending strength of 7.62 MPa at this point. Similarly, Figure 17a parameters are replaced into Formula (5) to generate the equivalent formula, and the partial derivative of x and y corresponding to formula z is determined. Solving the equations yields x = 0, y = 0.5, and z = 43.1. The glass fiber cement-based material reaches the maximum compressive strength of 43.1 MPa at this point. Figure 17b shows that the equation is solved to obtain x = 0, y = 1, and z = 6.7. The glass fiber cement-based material reaches the maximum bending strength of 6.7 MPa at this point. The parameters in Figure 18a are substituted into Formula (5) to obtain the corresponding formula, and then the partial derivative of x and y is calculated corresponding to the formula z. Solving the equations yields x = 0, y = 0.5, and z = 45.5. At this time, the PVA fiber cement-based material reaches the maximum compressive strength of 45.5 MPa. Similarly, as shown in Figure 18b, the equations are solved to obtain x = 0, y = 1, and z = 6.5. At this time, the PVA fiber cement-based material reaches the maximum bending strength of 7.62 MPa. The strength surface changes simulated by the finite element analysis results are consistent with the strength change surfaces simulated by the test results, proving that the finite element model can predict the mechanical properties well.

## 6. Conclusions

In this paper, the compressive, bending, and microscope observation tests of three fiber cement-based composite materials were mainly carried out by fiber volume fraction and freeze-thaw times. Its mechanical properties were analyzed under different volume fractions and freeze-thaw cycles, and its bivariate numerical model was established. The findings of finite element modeling were compared with actual testing using ABAQUS, and the failure mechanism was investigated.

1. Based on microscopic observation, the interface between carbon fiber and cement base is good, but the dispersion effect is not ideal. Glass fiber and PVA fiber are evenly dispersed and have better adaptability to the cement base, but the interface effect is relatively weak.

2. The compressive strength of the three fiber cement-based composites showed the law of first increasing and then decreasing with the volume fraction. For example, the highest compressive strength was 46.8 MPa for a 0.5% volume fraction of carbon fiber composite material. The bending strength follows a pattern of a quick rise initially, then gradually increasing as the volume fraction increases. The maximum bending strength was 7.6 MPa for a 1% carbon fiber composite. The compression test observed the frost resistance performance of carbon fiber > PVA fiber > glass fiber, and the bending test found the frost resistance performance of carbon fiber > glass fiber > PVA fiber.

3. The fiber’s frost resistance to the cement-based composite material has been much increased. As a result, the strength is much higher than the control group in both bending and compressive tests. Carbon fiber is better than PVA fiber, and PVA fiber is better than glass fiber. After 50 freeze-thaw cycles, the compressive strength of carbon fiber decreased by 3.8%, and the bending strength decreased by 3.9%. The compressive strength after 100 freeze-thaw cycles decreased by 9.6%, and the bending strength decreased by 7.9%.

4. The test findings demonstrate that the bending-compressive strength ratio increases as the fiber volume increases. As a result, fiber can significantly increase the sample’s toughness and ductility. However, the bending-compressive strength ratio has a different effect depending on the fiber. This discrepancy becomes more pronounced as the number of freeze-thaw cycles increases. Glass fiber is more significant than carbon fiber and PVA fiber.

5. According to the test results, a mathematical method for studying the bending strength of the three fibers under the combined action of different volume fractions and freeze-thaw cycles is proposed. This mathematical approach can accurately forecast all bending strengths in the range of 0–1% volume fraction and 0–100 freeze-thaw cycles, and it is also commonly used in engineering to obtain binary issue calculation answers.

6. Cloud images were simulated by analyzing ABAQUS. The fiber received very little force during compression, and the maximum stress at failure was 39 MPa. The fiber cement-based composite material was mostly shattered near the cement matrix’s bottom, with a maximum stress of 7.0 MPa. The limit load variation law of simulation and actual measurement results are relatively consistent, and the simulation results are slightly larger than the actual test values. The finite element model can accurately forecast bending and compressive strengths with a volume fraction greater than 1% and 100 freeze-thaw cycles.

## Figures and Tables

**Figure 1 materials-15-02226-f001:**
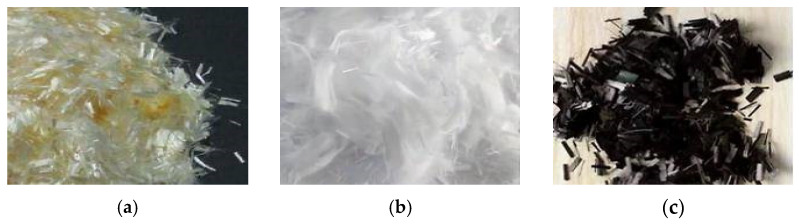
Test fibers: (**a**) PVA fiber; (**b**) Glass fiber; (**c**) Carbon fiber.

**Figure 2 materials-15-02226-f002:**
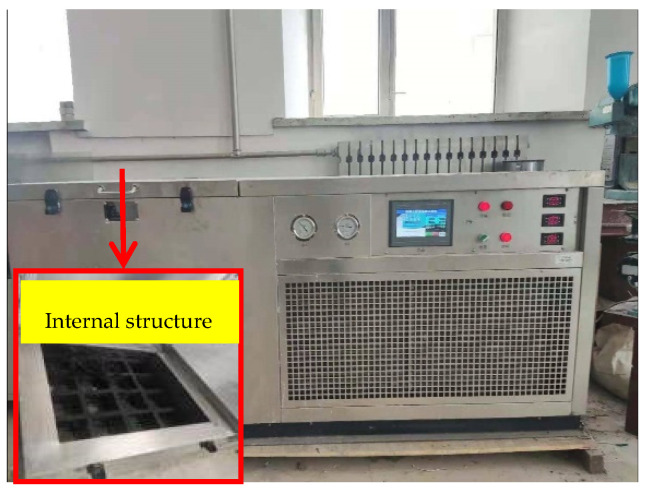
Freeze-thaw cycle machine.

**Figure 3 materials-15-02226-f003:**
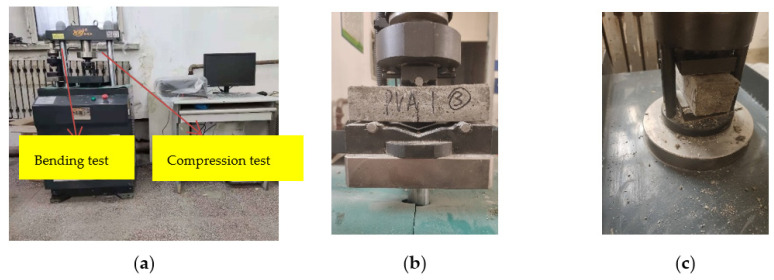
Mechanical test: (**a**) YAW-300H testing machine; (**b**) Bending test; (**c**) Compression test.

**Figure 4 materials-15-02226-f004:**
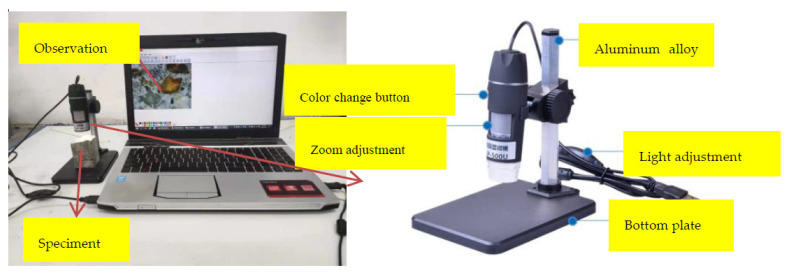
Schematic diagram of microscope observation.

**Figure 5 materials-15-02226-f005:**
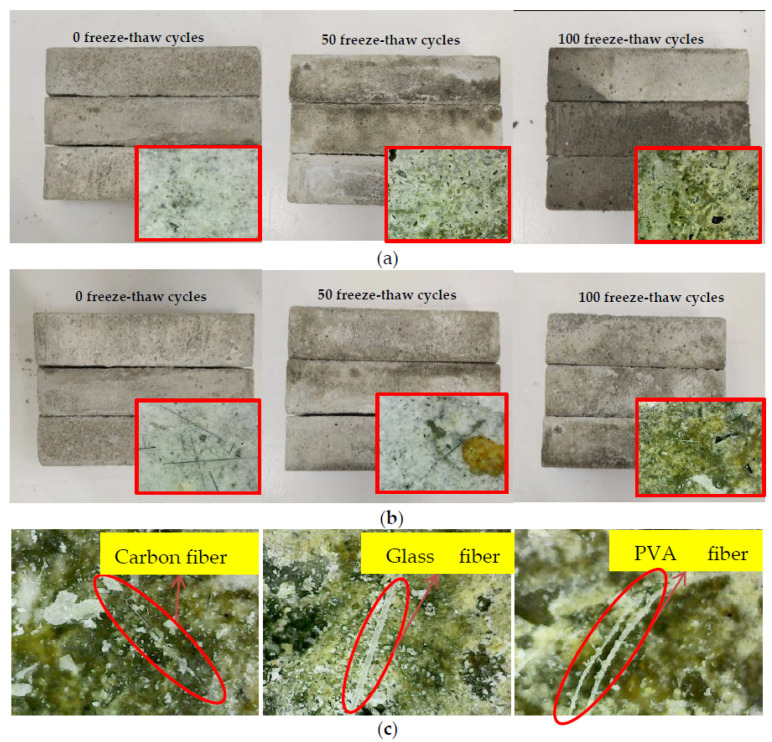
Surface condition of freeze-thaw specimen: (**a**) Without fiber; (**b**) With fiber; (**c**) Microscopic observation of different fibers.

**Figure 6 materials-15-02226-f006:**
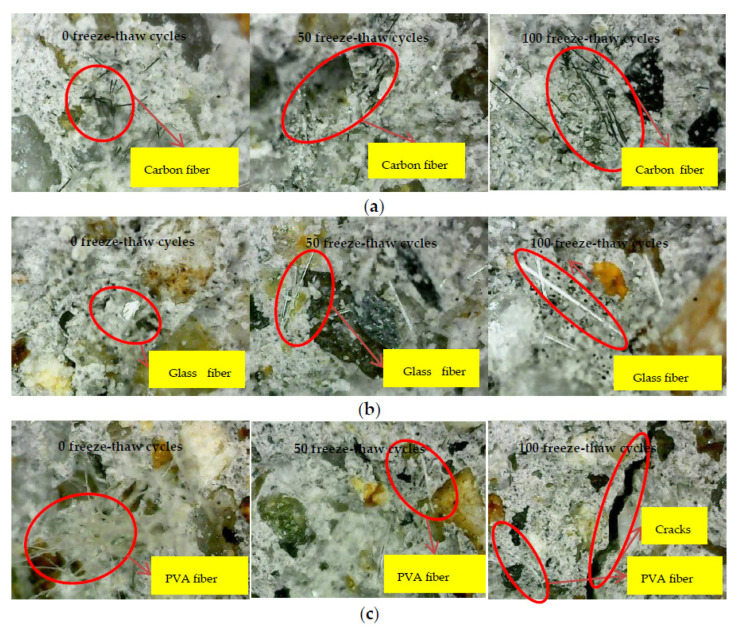
Microscopic observation: (**a**) Carbon fiber; (**b**) Glass fiber; (**c**) PVA fiber.

**Figure 7 materials-15-02226-f007:**
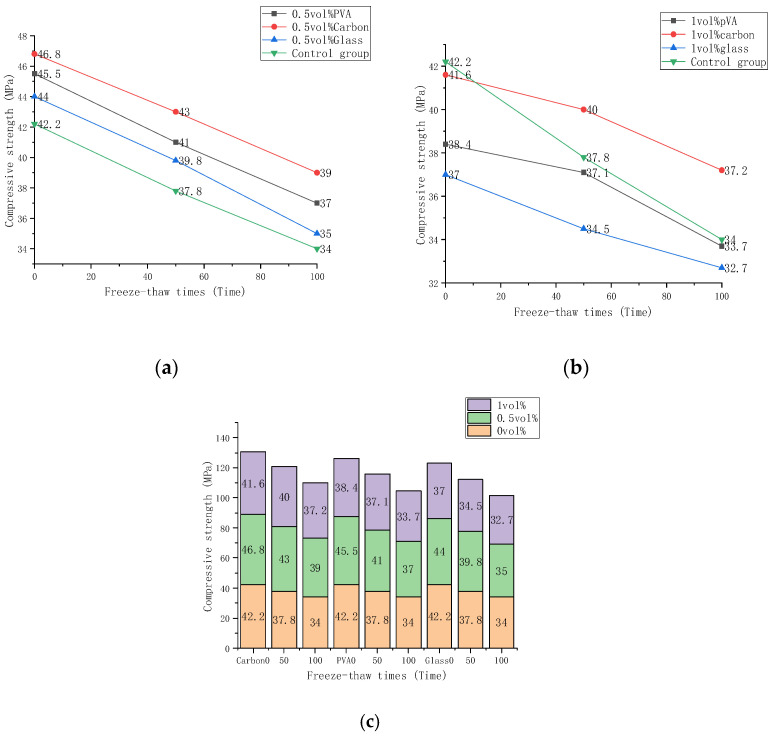
The compressive strength varies with the number of freeze-thaw cycles: (**a**) Compressive strength of 0.5 vol%; (**b**) Compressive strength of 1 vol%; (**c**) Compressive strength compared with 0 vol%, 0.5 vol%, 1 vol%.

**Figure 8 materials-15-02226-f008:**
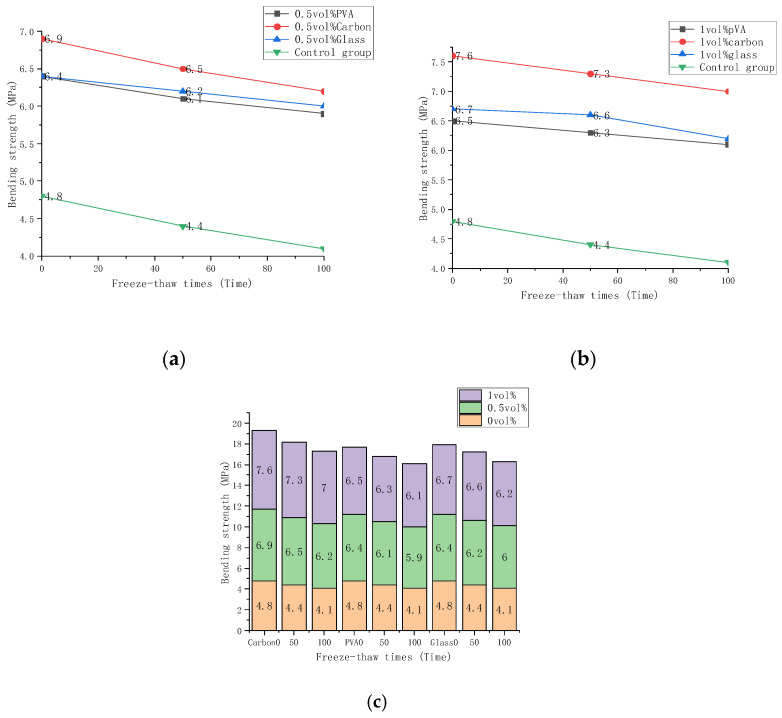
Bending strength changes with the number of freeze-thaw cycles: (**a**) Bending strength of 0.5 vol%; (**b**) Bending strength of 1 vol%; (**c**) The bending strength of 0 vol%, 0.5 vol%, 1 vol% comparison.

**Figure 9 materials-15-02226-f009:**
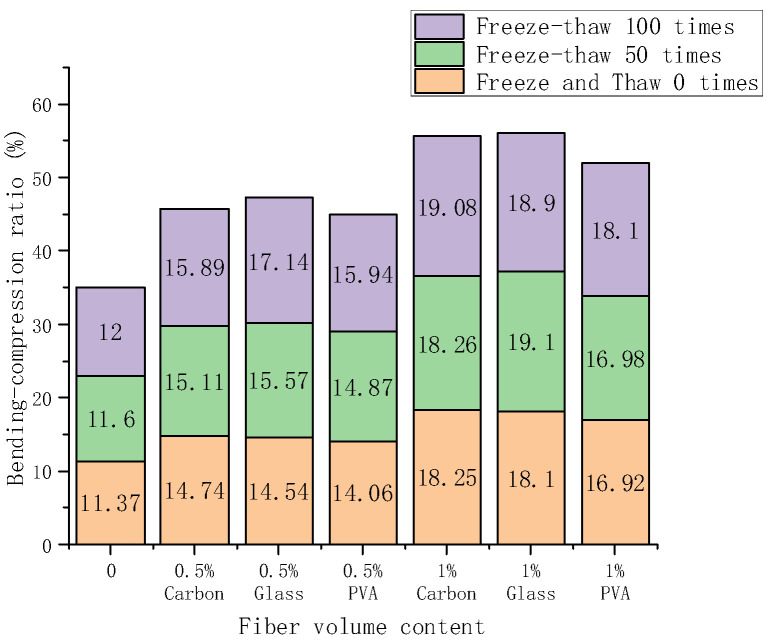
Variation of bending-compression ratio with doping and freeze-thaw time.

**Figure 10 materials-15-02226-f010:**
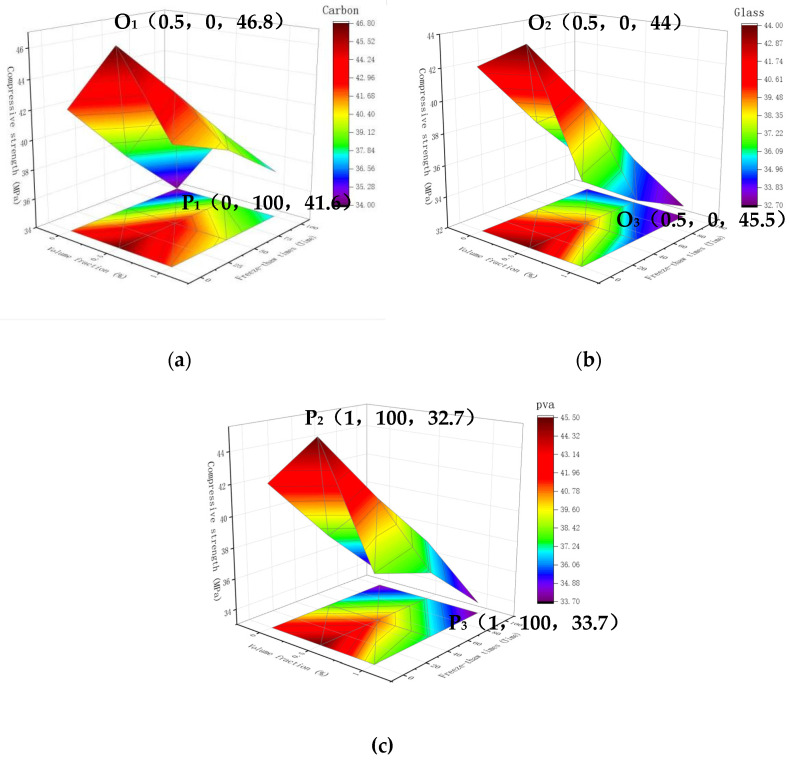
3D surface map with projection: (**a**) Carbon fiber compressive resistance; (**b**) Compression resistance of glass fiber; (**c**) PVA fiber compression resistance.

**Figure 11 materials-15-02226-f011:**
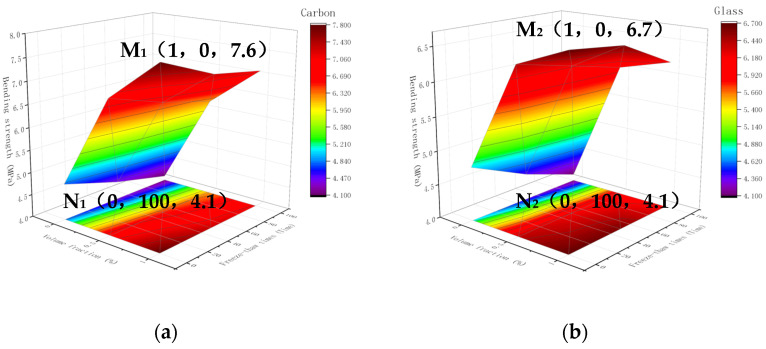
3D surface map with projection: (**a**) Carbon fiber bending resistance; (**b**) Glass fiber bending resistance; (**c**) PVA fiber bending resistance.

**Figure 12 materials-15-02226-f012:**
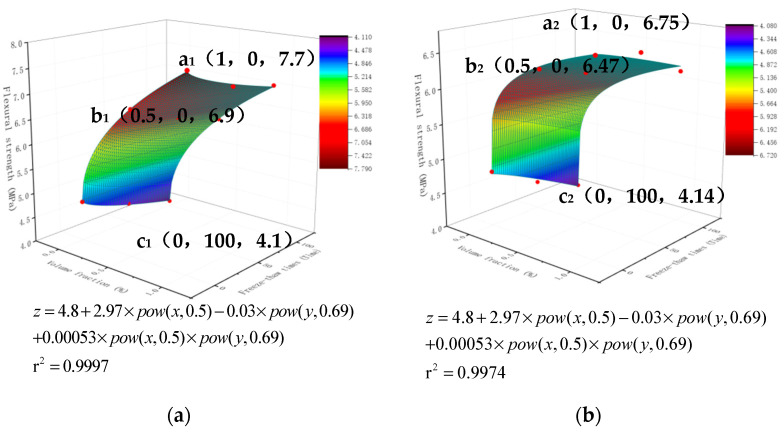
Origin Software Predicted Bending Strength Fitting Surface: (**a**) Carbon fiber; (**b**) Glass fiber; (**c**) PVA fiber.

**Figure 13 materials-15-02226-f013:**
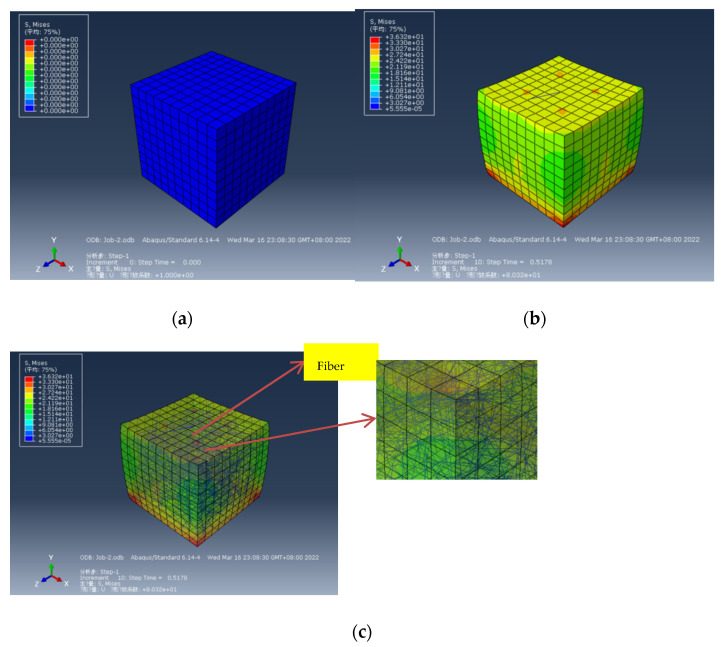
Simulation diagram of compression test: (**a**) Stress cloud diagram before force; (**b**) Stress cloud diagram after force; (**c**) ABAQUS simulated stress perspective cloud diagram.

**Figure 14 materials-15-02226-f014:**
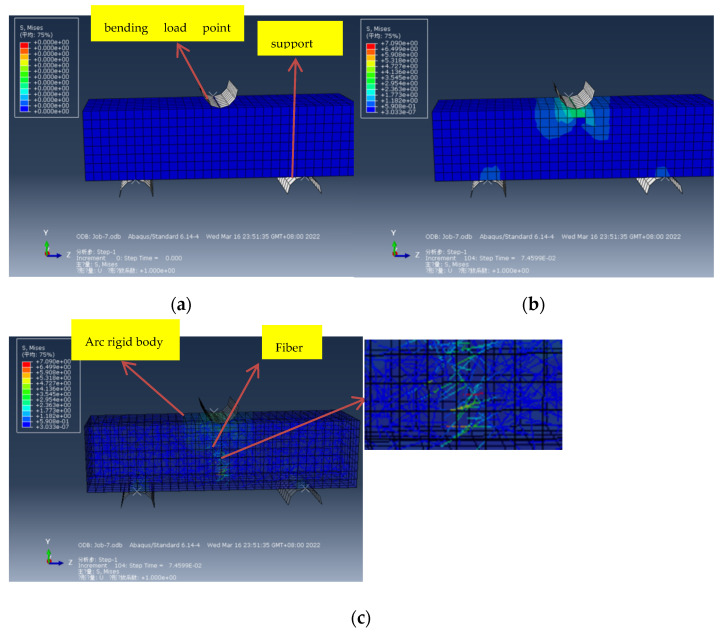
Bending resistance test: (**a**) Simulation stress cloud diagram before force; (**b**) Stress cloud diagram after force; (**c**) ABAQUS simulated stress perspective cloud diagram.

**Figure 15 materials-15-02226-f015:**
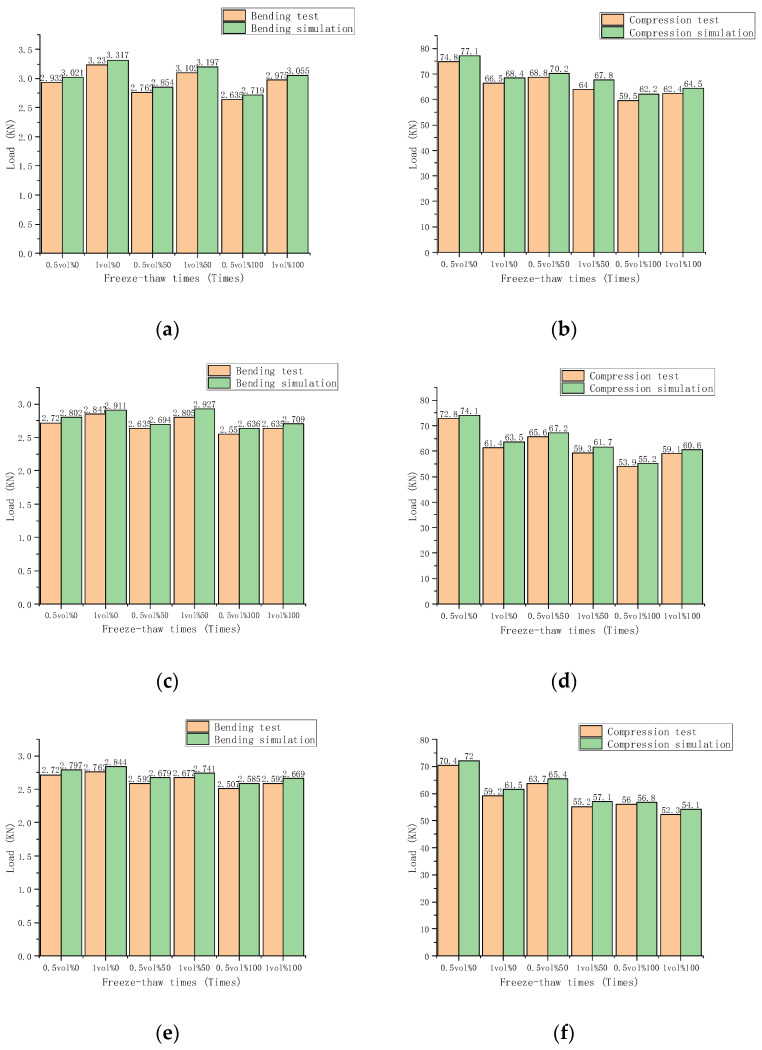
Comparison of fiber load: (**a**) Comparison of carbon fiber bending test; (**b**) Comparison of carbon fiber compression test; (**c**) Comparison of glass fiber bending resistance test; (**d**) Comparison of glass fiber compression test; (**e**) Comparison of PVA fiber bending resistance test; (**f**) Comparison of PVA fiber compression test.

**Figure 16 materials-15-02226-f016:**
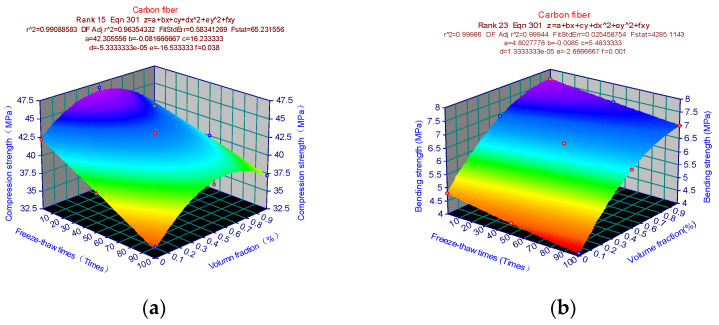
Carbon fiber strength prediction mode: (**a**) Compressive strength prediction model; (**b**) Bending strength prediction model.

**Figure 17 materials-15-02226-f017:**
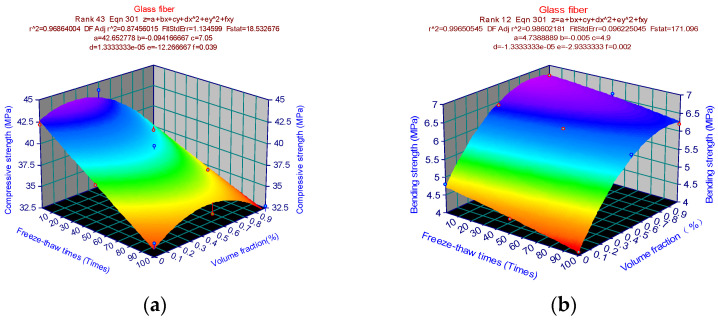
Glass fiber strength prediction mode: (**a**) Compressive strength prediction model; (**b**) Bending strength prediction model.

**Figure 18 materials-15-02226-f018:**
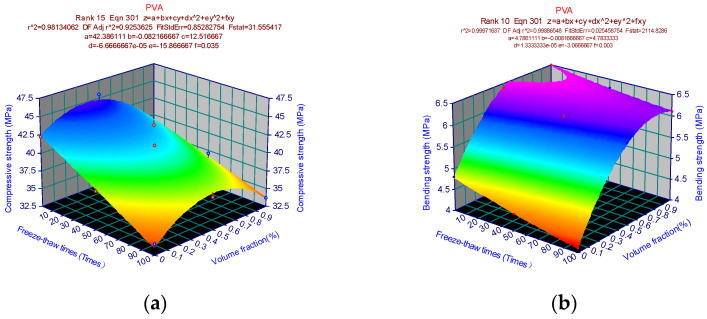
PVA fiber strength prediction mode: (**a**) Compressive strength prediction model; (**b**) Bending strength prediction model.

**Table 1 materials-15-02226-t001:** Material property of cement.

Fineness Modulus	Water Requirement of Normal Consistency/%	Stabilities	Setting Time/min		Bending Strength/MPa		Compressive Strength/MPa	
			Initial Setting	Final Setting	3 d	28 d	3 d	28 d
3.2	25.4	18	160	210	5.6	9.4	25.8	45.2

**Table 2 materials-15-02226-t002:** Particle size gradation.

Square Hole Sieve Diameter/mm	2.00	1.60	1.00	0.50	0.16	0.08
Cumulative sieve/%	0	7 ± 5	33 ± 5	67 ± 5	87 ± 5	99 ± 5

**Table 3 materials-15-02226-t003:** Fiber performance indicators.

Material Type	Tensile Strength/MPa	Elongation/%	Fiber Density/(g/cm^3^)	Elastic Modulus/GPa
Carbon fiber	3500	1.5	1.6	230
Glass fiber	2500	3.6	4.8	70
PVA fiber	1900	8	1.3	35

**Table 4 materials-15-02226-t004:** Concrete mix design.

NO.	Cement /(kg/m^3^)	Sand /(kg/m^3^)	Water /(kg/m^3^)	Water Cement Ratio	Carbon Fiber /(kg/m^3^)	Glass Fiber /(kg/m^3^)	PVA Fiber /(kg/m^3^)	Superplasticizer /(kg/m^3^)
1	700	1400	350	0.5	0(0%)	0(0%)	0(0%)	0
2	700	1400	350	0.5	0.8(0.5%)	0(0%)	0(0%)	5
3	700	1400	350	0.5	1.6(1%)	0(0%)	0(0%)	15
4	700	1400	350	0.5	0(0%)	2.4(0.5%)	0(0%)	5
5	700	1400	350	0.5	0(0%)	4.8(1%)	0(0%)	15
6	700	1400	350	0.5	0(0%)	0(0%)	0.65(0.5%)	5
7	700	1400	350	0.5	0(0%)	0(0%)	1.3(1%)	15

**Table 5 materials-15-02226-t005:** Compressive strength.

Material Type	Volume Fraction (%)	Freeze-Thaw Times	Compressive Strength (Mpa)					
			1	2	3	Average Value	Standard Deviation	Salience
Control group	0	0	41.6	42.5	42.5	42.2	0.52	3.3208 × 10^−20^
	0	50	37	38.2	38.2	37.8	0.7	
	0	100	33.1	33.6	35.3	34	1.15	
Carbon fiber	0.5	0	45.7	47.1	47.5	46.8	0.95	
	0.5	50	42.3	42.6	44.1	43	0.96	
	0.5	100	37.1	39.2	40.7	39	1.81	
	1	0	40.5	41.4	42.9	41.6	1.21	
	1	50	38.9	39.5	41.6	40	1.42	
	1	100	35.9	38.7	37	37.2	1.41	
Glass fiber	0.5	0	44.7	43.5	43.8	44	0.62	
	0.5	50	39.8	38.8	40.8	39.8	1	
	0.5	100	35.7	34.4	33.4	34.5	1.15	
	1	0	36.0	37.9	37.1	37	0.95	
	1	50	35.9	33.4	34.2	34.5	1.28	
	1	100	34.6	32	31.5	32.7	1.67	
PVA fiber	0.5	0	45.6	46.1	44.8	45.5	0.66	
	0.5	50	40.7	40.3	42	41	0.89	
	0.5	100	36.5	38.3	36.2	37	1.14	
	1	0	39	37.2	39	38.4	1.04	
	1	50	38.5	36.3	36.5	37.1	1.22	
	1	100	34.4	32.1	34.6	33.7	1.39	

**Table 6 materials-15-02226-t006:** Bending strength.

Material Type	Volume Fraction (%)	Freeze-Thaw Times	Bending Strength (Mpa)					
			1	2	3	Average Value	Standard Deviation	Salience
Control group	0	0	4.8	4.9	4.7	4.8	0.1	1.5981 × 10^−13^
	0	50	4.3	4.6	4.3	4.4	0.17	
	0	100	4.3	4.1	3.9	4.1	0.2	
Carbon fiber	0.5	0	6.7	6.8	7.2	6.9	0.26	
	0.5	50	6.8	6.5	6.2	6.5	0.3	
	0.5	100	5.9	6.2	6.5	6.2	0.3	
	1	0	7.8	7.8	7.2	7.6	0.35	
	1	50	7.6	7.5	6.8	7.3	0.44	
	1	100	6.8	7	7.2	7	0.2	
Glass fiber	0.5	0	6.5	6.4	6.3	6.4	0.1	
	0.5	50	6.2	6.4	6	6.2	0.2	
	0.5	100	6.8	5.6	5.6	6	0.69	
	1	0	6.7	6.9	6.5	6.7	0.2	
	1	50	6.9	7.0	5.9	6.6	0.6	
	1	100	6.5	6.4	5.7	6.2	0.44	
PVA fiber	0.5	0	6.5	6.5	6.2	6.4	0.17	
	0.5	50	5.8	6.5	6	6.1	0.36	
	0.5	100	5.8	6.4	5.5	5.9	0.46	
	1	0	6.6	6.9	6	6.5	0.46	
	1	50	6.3	6.3	6.3	6.3	0	
	1	100	6.8	6.2	5.3	6.1	0.75	

**Table 7 materials-15-02226-t007:** The bending-compressive ratio.

Material Type	Volume Fraction	The Bending-Compression Ratio (%)		
		Freeze and Thaw 0 times	Freeze and Thaw 50 times	Freeze and Thaw 100 times
Control group	0 vol%	11.37	11.6	12
Carbon fiber	0.5 vol%	14.74	15.11	15.89
	1 vol%	18.25	18.26	19.08
Glass fiber	0.5 vol%	14.54	15.57	17.14
	1 vol%	18.1	19.1	18.9
PVA fiber	0.5 vol%	14.06	14.87	15.94
	1 vol%	16.92	16.98	18.1

**Table 8 materials-15-02226-t008:** Different fiber parameters.

Fiber Type	z_0_	B	C	D	E	F
Carbon fiber	4.80	2.97	0.50	–0.03	0.69	0.00053
Glass fiber	4.80	1.95	0.22	–0.004	1.107	0.0015
PVA fiber	4.80	1.73	0.14	–0.02	0.82	0.006

**Table 9 materials-15-02226-t009:** Pearson correlation coefficient of carbon fiber cement-based materials.

Carbon Fiber	Compressive Strength	Bending Strength	Freeze-Thaw Times	Volume Fraction
Compressive strength	1			
Bending strength	0.490	1		
Freeze-thaw times	−0.787 *	−0.219	1	
Volume fraction	0.185	0.942 **	0.000	1

* At the 0.05 level (two-tailed), the correlation is significant. ** At the 0.01 level (two-tailed), the correlation is significant.

**Table 10 materials-15-02226-t010:** Pearson correlation coefficient of glass fiber cement-based materials.

Glass Fiber	Compressive Strength	Bending Strength	Freeze-Thaw Times	Volume Fraction
Compressive strength	1			
Bending strength	0.006	1		
Freeze-thaw times	−0.809 **	−0.232	1	
Volume fraction	−0.361	0.898 **	0.000	1

** At the 0.01 level (two-tailed), the correlation is significant.

**Table 11 materials-15-02226-t011:** Pearson correlation coefficient of PVA fiber cement-based materials.

PVA Fiber	Compressive Strength	Bending Strength	Freeze-Thaw Times	Volume Fraction
Compressive strength	1			
Bending strength	0.267	1		
Freeze-thaw times	−0.808 **	−0.249	1	
Volume fraction	−0.181	0.873 **	0.000	1

** At the 0.01 level (two-tailed), the correlation is significant.

## Data Availability

All data generated or analyzed during this study are included in this published article.

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
