# Peer review of "Frost Resistance Investigation of Fiber-Doped Cementitious Composites"

_materials, 2022, doi:10.3390/ma15062226_

Round 1

Reviewer 1 Report

This study investigated the frost resistance effect of fibers content in cementitious materials. The flexural strength performance of three cement-based materials including carbon fiber- , glass fiber- and PVA fiber cement-based composites based on the freeze-thaw effect simulation 

There are many comments should be addressed as follows:

1- in the abstract part

i- The author should emphasize and highlight the main objective and challenge, and therefore the first two sentences should be reviewed
ii-  The novelty and approach must be clearly clarified
iii- The main result should be clearly presented

2- The introduction section is good but there are some changes needed as follows:
a- clear information related to the goal, the problem of the current study, the existing challenges, 
b- The advantage of the current study
c- The author should present a clear claim at the end of this introduction to cover the objective, technology used, innovation and key findings

3- regarding the Results and discussion part 
a- the material composition should be confirmed for example XRD and  EDS analysis can be used
b- the experimental tensile strength values of tested specimens should be clarified at all ratio used such as 0%, 0.5%, and 1%
c- The author should provide the reason to select , 0.5vol%, 1vol%, is it possible to increase this value?

d- The figure captions is not enough to describe the figure, the author must provide more information and details enough to clarify the figures.

e- It is recommended to change the subtitle of "5 summary" to be "5 conclusion".

 Moreover, this section is long and recommended to be reviewed and rewritten as one paragraph in the bases of the objective of the present study and the main findings obtained, feature.

Author Response

Dear Reviewer,

The authors would like to thank the reviewers for their review of the manuscript and valuable suggestions. All comments have been carefully considered and accounted for in a revised version of the manuscript.

Regards,

Wei Li

Reviewer 2 Report

The paper is devoted to the influence of temperature change (freezing and thawing) on mechanical properties of concrete reinforced by threads of carbon, glass or PVA. The paper consists of the experimental part and numerical part where the results were compared to each other. It begins with an introduction that describes investigations on composite reinforced concrete, but no work devoted to freeze and thaw of composites is mentioned.

The experimental part is constructed in such a way as to hide all important details of the research. The details of experiments are given as a reference to the Chinese standards which hardly to obtain to the reader out of China. In different parts of the paper different specimen’s dimensions for the same specimens are given. Therefore, the typical reader is not even imagining how the experiments have been performed. The photos are low resolution and are difficult to see details. Finally, no single result is presented! Probably the mean values are given, but it is even not declared. There is no statistical analysis of the results. The groups of different percentages of reinforced and investigated in different temperature conditions are not statistically compared.

Microscopic observations described in the paper are not important for the subject of the research. They can probably define the correctness of specimen preparation. But also here no details, which parts of the specimens are investigated cannot be found.

The mathematical decompositions of the obtained laboratory results are made for arbitrarily chosen functions and there is no explanation, why this type of function has been chosen.

The construction of the numerical model is not given and there is no reference where it can be found.

In the chapter on correlation analysis (what will be compared?) the tables 8-10 are completely incomprehensible.

The detailed remarks are indicated in the paper text.

Author Response

(The authors gave the same response as above.)

Reviewer 3 Report

  • In the Introduction Section, the authors should include previous studies on the effect of fibers on the F-T resistance of cement composites.
  • The characteristics of the aggregates mentioned in Section 2.1 should be provided: what is the particle size gradation?
  • Did the authors use chemical admixtures to control the workability of concrete containing 1% fiber? Was air-entraining agent used? What is the air content of concrete mixtures?
  • What the authors are trying to say in Lines 218-219 is confusing.
  • The information in Section 3.1 and 3.2 are quite similar. So, it might be better to combine the two sections.
  • The authors should discuss the mechanisms behind the improvements in the compressive strength of the fiber reinforced concretes. Why did some fibers perform better than the others?
  • In Lines 324-327, there is no connection between these three sentences “Therefore, the frost resistance in the compression test is carbon fiber> PVA fiber> glass fiber. Analyze the reason for the physical performance index. The elastic modulus is the largest, which is reflected by the analysis of the microscope observation results.” What the authors are trying to say is unclear.
  • The authors reported that carbon fiber reinforced concrete had the flexural strength loss on exposure. However, no discussion was presented about the reasons for the superior F-T resistance of carbon fiber reinforced concrete specimen.
  • In Lines 409-410, the authors stated “The fiber volume is positively related to the more excellent compression ratio. As a result, fiber can significantly increase the sample's toughness and ductility.” However, from the literature, fibers are generally added to cement-based materials to increase flexural toughness/tensile strength properties and not compression strength. The authors should explain why in this study, the fiber volume is more positively correlated with the compressive strength.
  • Relative dynamic modulus of elasticity of concrete is usually the main evaluation criteria for the F-T resistance. Why did the authors fail to include this test in the study?

Author Response

Dear Reviewer,

The authors would like to thank the reviewers for their review of the manuscript and valuable suggestions. All comments have been carefully considered and accounted for in a new version of the manuscript.

Yongcheng Ji

Round 2

Reviewer 2 Report

The current version of the paper contains many important corrections e.g. much better is experiment description, some language errors have been removed. Unfortunately, not all remarks made by both reviewers to the previous version have been applied in fact. Here is the list of main drawbacks of the current version:

  1. In the opinion of version No. 1, there was no request to change ALL “flexural” to “bending”. In fact, there was a suggestion to make such a change. Unfortunately, the Authors made this change automatically, not thinking about the meaning of each change. As the result, they have changed the titles of two papers in the literature [18] and [24]!! On the other hand, the figures containing this word (Fig. 8a,b, Fig. 9, Fig. 11a,b,c, Fig. 12a,b,c, Fig. 17b, Fig. 18b) have been not corrected.
  2. The current version of the paper still does not contain the statistical analysis of the results. There is no proof (what is stated in the Authors response) that the changes of properties are statistically important. As the results of single tests still are not presented it is not possible to perform for example the ANOVA analysis, which is a statistical tool to confirm such statement.
  3. According to new details included in the current version, the specimens for the compression tests have been cut from the remains of bending tests specimens. It is a serious mistake. After the bending test until the rapture, the properties in the volume of both parts of the broken specimen can be different than in the virgin one (due e.g. because of not visible microcracks). Therefore, all results of compression tests are questionable. Was it difficult to produce additional virgin specimens for compression tests? Four such specimens could be cut out from the single bending test specimen.
  4. How the effective size of the compression specimens can be “0 mm(length) x 40 mm (width) x 40 mm (height)”? According to Fig. 3c and Fig. 13, they were rather cubic 40x40x40 mm.

Also, the bending specimen dimensions are still wrong. Should be: 40 mm (height) × 40 mm (width) × 160 mm (length).

  1. Tables included in Fig. 12 a,b,c are not easily readable and contain Chinese descriptions.
  2. Tables 8-10 are still not enough explained.

Concluding, I still can not recommend the manuscript for publication especially since some serious mistakes in the planning of the experiment has been detected.

Author Response

Dear Reviewer,

The authors would like to thank the reviewers for their review of the manuscript and valuable suggestions. All comments have been carefully considered and accounted for in a revised version of the manuscript. 

Regards,

Yongcheng Ji

Reviewer 3 Report

Although the authors did revise the manuscript, I have also made some additional editing as shown below:

Lines 218-219

Partial scaling of the concrete surface after 100 freeze-thaw cycles. However, the addition of the fibers prevented scaling damage thereby preserving the specimen's integrity.”

Lines 324-327

 ‘Carbon fibers have a high modulus of elasticity (Table 3), and relatively minimal mortar slippage was observed by microscopy. Therefore, based on compression test and microscopy observations, it seems that the frost resistance of fiber reinforced concrete mixtures evolved according to this sequence; carbon fiber > PVA fiber > glass fiber.”

Author Response

Dear Reviewer,

The authors would like to thank the reviewers for their review of the manuscript and valuable suggestions. All comments have been carefully considered and accounted for in a new version of the manuscript.

Yours truly,

Wei Li
